# Analysis of the Physicochemical, Mechanical, and Electrochemical Parameters and Their Impact on the Internal and External SCC of Carbon Steel Pipelines

**DOI:** 10.3390/ma13245771

**Published:** 2020-12-17

**Authors:** Luis Manuel Quej-Ake, Jesús Noé Rivera-Olvera, Yureel del Rosario Domínguez-Aguilar, Itzel Ariadna Avelino-Jiménez, Vicente Garibay-Febles, Icoquih Zapata-Peñasco

**Affiliations:** 1Instituto Mexicano del Petróleo, Eje Central Lázaro Cárdenas Norte 152, San Bartolo Atepehuacan, Alcaldía Gustavo A. Madero, Ciudad de México C.P.07730, Mexico; yureelrda@gmail.com (Y.d.R.D.-A.); iaavelino@imp.mx (I.A.A.-J.); vgaribay@imp.mx (V.G.-F.); 2Tecnológico Nacional de México, Campus Ixtapaluca, TESI, Km. 7 de la Carretera Ixtapaluca-Coatepec s/n, Ixtapaluca, Estado de México C.P.56580, Mexico; jnoe.rivera@tesi.edu.mx

**Keywords:** stress corrosion cracking (SCC), external-SCC, internal-SCC, pipeline steel

## Abstract

The review presented herein is regarding the stress corrosion cracking (SCC) phenomena of carbon steel pipelines affected by the corrosive electrolytes that comes from external (E) and internal (I) environments, as well as the susceptibility and tensile stress on the SCC. Some useful tools are presented including essential aspects for determining and describing the E-SCC and I-SCC in oil and gas pipelines. Therefore, this study aims to present a comprehensive and critical review of a brief experimental summary, and a comparison of physicochemical, mechanical, and electrochemical data affecting external and internal SCC in carbon steel pipelines exposed to corrosive media have been conducted. The SCC, hydrogen-induced cracking (HIC), hydrogen embrittlement, and sulfide stress cracking (SSC) are attributed to the pH, and to hydrogen becoming more corrosive by combining external and internal sources promoting cracking, such as sulfide compounds, acidic soils, acidic atmospheric compounds, hydrochloric acid, sulfuric acid, sodium hydroxide, organic acids (acetic acid, mainly), bacteria induced corrosion, cathodic polarization, among others. SCC growth is a reaction between the microstructural, chemical, and mechanical effects and it depends on the external and internal environmental sources promoting unpredictable cracks and fractures. In some cases, E-SCC could be initiated by hydrogen that comes from the over-voltage during the cathodic protection processes. I-SCC could be activated by over-operating pressure and temperature at flowing media during the production, gathering, storage and transportation of wet hydrocarbons through pipelines. The mechanical properties related to I-SCC were higher in comparison with those reviewed by E-SCC, suggesting that pipelines suffer more susceptibility to I-SCC. When a pipeline is designed, the internal fluid being transported (changes of environments) and the external environment concerning SCC should be considered. This review offers a good starting point for newcomers into the field, it is written as a tutorial, and covers a large number of basic standards in the area.

## 1. Introduction

In upstream and downstream oil and gas industries two major problems are found during operating pipeline system transporting hydrocarbons that are related to external and internal corrosion damage, where corrosion surfaces are fundamentally caused by water containing aggressive ions as the dominant parameter. It is not surprising that a mechanical failure could be started by water solution as well [1,2,3]. It can become somewhat significant when the change of strain, shape, and damage are not subjected to control by the external or internal applied force on pipelines. The external and internal corrosion processes have negative effects which in turn both generalized (uniform) and localized corrosion damage could be attributed to stress corrosion cracking (SCC) [4,5,6,7,8,9,10,11,12,13]. They could spread in a wide range of systems from the pipeline walls. Common solutions to this are cathodic protection, coatings, and chemical treatment applications. The negative effects of the phenomena of external SCC on carbon steel pipelines could be influenced by the hydrogen generated during cathodic protection, prepared soil solutions, carbonates and bicarbonate solutions as well as the hydrogen that originate from the sulfide compounds, HIC from acidic soils, acidic atmospheric compounds, organic acids, external bacteria induced corrosion, among other external solution promoting cracking [14,15,16,17,18,19,20,21,22,23,24]. Oxygen and hydrogen that comes from the low and high (overprotection) cathodic protection, respectively, becoming negative effects for external SCC [25,26]. Negative effects required for the occurrence of internal SCC are attributed to aggressive gas and pH that comes from oilfield produced water, sour media or brine sour media (hydrogen sulfide compounds, bisulfide compounds, and carbon dioxide), sulfuric acid, naphthenic acids, hydrochloric acid, organic acids (acetic acid, mainly), sodium hydroxide, internal bacteria induced corrosion, cathodic polarization, over-operating pressure, elevated temperatures, among other data of internal environment promoting cracking [13,27,28,29,30,31,32]. Others are related to the incorrect inhibitor types complimented with an inappropriate operating pressure where a low inhibition efficiency could be achieved [13,33]. As a result, other special forms of external and internal corrosion such as hydrogen blistering, hydrogen embrittlement, intergranular corrosion, pitting corrosion, microbiological corrosion (aerobic or anaerobic), corrosion under deposits, cell concentrations, galvanic corrosion near to weld, erosion, coating failures, among other mechanisms can be identified [34,35,36,37,38,39]. This is because the carbon steel containing high concentration of iron tends to return to the native ores in the form of oxide or hydroxide compounds (native compounds) after its refining process in a blast furnace. It is important to mention that the pipelines are usually designed, constructed and installed in consideration of the factors of the external and internal environments that they are exposed to, including metallurgy and stresses (including residual, applied, and criteria safety factors) [40,41]. However, the SCC growth is a type of unpredictable corrosion damage where a static tensile and constant load (mechanical effects) for pipelines exposed to corrosive environments is applied [1,42]. Consequently, the original mechanical properties of the carbon steel pipelines may be modified (displaced) toward high susceptibility to SSC (poor mechanical properties), where the crack and fracture starts [11]. Then, cracks propagate within deformations to achieve a total failure, and this should be tested at laboratory level (experimental scenarios before and after electrochemical evaluations) by combining electrochemical and mechanical measurements [43,44]. For that reason, the range of variables studied and reported in the literature is especially important. In the present work, a brief review regarding the SCC phenomena data is discussed. Then the corrosive electrolytes and other parameters reported in the literature regarding the external and internal SCC of carbon steel pipelines based on physicochemical, mechanical, and electrochemical data are highlighted. An experimental review, a brief experimental summary, and a comparison for physicochemical, mechanical and electrochemical data affecting external and internal SCC for carbon steels exposed to external and internal corrosive media were carried out.

## 2. A Review of the SCC Phenomena

Stress corrosion cracking (SCC) is a phenomenon by which a material cracks because of the action of a corrosive media and tensile stresses into the material. External-stress corrosion cracking (E-SCC) is a cause of failure in buried pipelines transporting hydrocarbons and its derivatives where the oxides start from the outside diameter (OD) of the pipeline. The external protective coating can suffer damage, and a wet environment may be generated between the coating and pipeline surface as a function of time, and this environment may induce the development of E-SCC degrading it. Internal stress corrosion cracking (I-SCC) in pipelines may be produced by internal fluid that contain different partial pressure of acid sulfide (H_2_S), carbon dioxide (CO_2_), carbonic acid (H_2_CO_3_), naphthenic acid, oxygen, and water (H_2_O). The corrosion problem described before is increased by the minimum or maximum operating pressure and temperature. E-SCC can be found on gas pipeline more than in liquid pipeline systems, as was reported by Afanasyev et al. [45]. The authors revealed that the variables affecting the distribution of SCC could be attributed to the low resistivity (about 100 ohm.cm) of the soil brine, prolonged exposure time (up to 20 years of operation in real conditions), location, pipeline inclination (degree), elevation profile, large diameter, long pipeline network (pipeline sections: around length of 25 km), large diameter of the pipeline, random operating pressure, and temperature. As a result, significant degraded coating could be achieved where growth in depth and opening width (wide variation of lengths and widths), and possible anomalies-defects-indications (defect population grown for 20 years) must be taken into consideration. It can lead to induced crack or fracture propagation in E-SCC. Crack length-to-width ratio and width-to-depth ratio may be studied using a cyclic test simulating a constant load similar to the operational load. In addition, they concluded that the proposed methodology can be applied when actual samples are correlated with the inline inspection technology (ILI) measurements to combat the negative effects supporting the ILI data analysis and avoid hard choices (including excavations) during the validation of prioritization in the integrity assessment [46]. It is overly important to mention that external loads (natural or applied loads including gravity and weight) presence could also affect the coated pipeline from extended sections. For that reason, damaged coating and SCC occurrence may be achieved in large diameter steel pipelines.

### 2.1. When SCC Occur

Figure 1 shows the main factors required for guideline E-SCC and I-SCC to be presented in a metallic material.

The presence of the SCC is not always visible (visual inspection in field experiences), but this defect is dangerous, especially when it is found like colonies of cracks, which coalesce as the time evolves to cause ruptures. It is a phenomenon that can happen unexpectedly and quickly after a satisfactory period of service, leading to a catastrophic failure in pipelines. The main variables that have an effect on the SCC from OD and inside diameter (ID) are:


**OD**
Water;Environment;pH in soil;Electrolytes in soil;Steel grade;Chemical composition;Microstructure;Hardness;Toughness;Elasticity;Elongation;Fatigue;Type of welding;Cathodic protection;Type of coating;Microorganisms.



**ID**
Water;Electrolytes in water (oilfield produced water);pH in solution;Hydrocarbon transported;Chemical treatment application;Pressure;Temperature;Flowing media;Microorganisms;Internal coating.


### 2.2. Origins of Stresses

Cheng [1] reported that the interaction between metallurgy, aggressive environment, and the applied or residual stress on operating pipelines coexist. The residual stress refers to a natural stress state, which is held in a metal in the absence of external load, gravity, thermal gradient, among other source of stresses [47]. Applied stress come from pressure-microstructure manufacturing. It could occur from plastic deformation (rolling, extruding, bending, and forming), elastic deflection, manufacturing process (welding, machining, and electrodeposition), heat or thermochemical treatment (quenching and ion plating) [48,49,50,51].

#### 2.2.1. Microstructure

Data acquired from the interaction-reaction effects depend on the specific environment promoting the cracks or fractures occurrences. In the case of metallurgy this is related to susceptible sites such as microstructural features (chemistry, alloying elements, texture, grain size, shape distribution, secondary phases, random, special grain boundaries, and intergranular degradation) becoming anodic affecting the susceptibility to SCC at certain type of cracks and fractures [1,2,52]. In some cases, the SCC problem appears to be a poor manufacturing process for carbon steel pipelines. This problem could be increased when incorrect finished surface may occur, where the expected susceptibility to corrosion, SCC occurrence, and failure frequency are a complicated process. However, microstructural features are strongly affected from the corrosiveness of the environment. Thus, active electrochemical reactions are expected on cracks-fractures/test solution interfaces where the values for mechanical features for steels go up increases by using a standard tensile test [8,44,53]. The interactions between the physicochemical, mechanical, and electrochemical factors depend on many variables, such as changes of the corrosive media and residual or applied stresses resulting in several forms of corrosion and microstructural damage (microstructural degradation) [34,35]. SCC may be included in the HIC, hydrogen embrittlement, and SSC that comes from the testing of steels in hydrogen sources from the over-voltage during the cathodic protection, representative soil solutions, carbonates, bicarbonates, acidic soils, acidic atmospheric compounds, organic acids, external bacteria induced corrosion, among other data [1,2,3,34,35].

#### 2.2.2. Testing

The mechanical testing may be addressed by four stages to study the crack growth rate in SCC mechanisms using slow strain rate tests (SSRT) or constant load (CL) such as (a) initiation conditions develop (nucleation), (b) cracks initiate with slow growth, (c) growth by continuous initiation, extension, and coalescence, and (d) large cracks coalescence with propagation and failure [1,54].

### 2.3. Techniques Used for Assessing SCC

Perhaps the three techniques most used to assess SCC are (1) slow strain rate tests (SSRT) using constant extension rate machine, (2) constant load tests (CLT) using proofs rings and (3) small punch tests (SPT) using miniaturized specimens. Sometimes these techniques are combined with electrochemical measurements in order to monitoring the SCC process.

The SSRT is a mechanical technique for studying cracking environments of buried pipelines. It is widely applied for E-SCC and I-SCC evaluations of carbon steels pipelines. The SSRT technique involves the slow straining of a specimen of the steel of interest in a solution in which will be in service. This technique provides useful information on SCC susceptibility of the actual pipeline steels exposed to external and internal solutions, in addition to a relatively short experimental time to evaluate SCC susceptibility of metal materials [55,56,57,58]. There are several mobile constant extension rate machines used in the SSRT to determine SCC susceptibility of metals exposed to corrosive environments [59,60].

In addition, various types of proof rings are used to evaluate SCC a constant load (©2017 CORTEST and all work represented. All rights reserved). These tests are performed generally at higher stress than yielding strength. The samples are exposed to a corrosive environment for 30 days and the criterion applied is pass/no pass; if material did not failed pass and not pass if material failed. These tests are carried out according to ANSI/NACE TM0177 [61]. Some limitations of proof rings are that is a static test, evaluate the elastic zone, material could not fail, relaxation could occur over the time, the SCC process is not monitored, the test is performed at only determined stress, long time of tests, among others.

To evaluate some mechanical properties of metal materials there is a small punch test (SPT) [62,63]. It consists of evaluating susceptibility to SCC using a small punch and acoustic emission (AE). Typically, 10.0 × 10.0 × 0.5 mm specimens are tested at various deformation rates; the amplitude of the AE signal is captured and the equivalent of fracture deformation and susceptibility to SCC of high-strength steels is taken. Because of the small size of the specimens that the punch test uses, it is an appropriate experiment for mechanically characterizing small areas, which it would not be possible to analyze with the other two types of tests. The SPT assay is currently being used to determine the mechanical properties related to axial and biaxial state of tension in the small punch test, ductile-brittle transition temperature, the fracture and creep behavior, as well as a study related to the SCC susceptibility of metals to environmentally assisted cracking (EAC) [53].

#### SCC Assessment Using SSRT

SCC susceptibility is commonly tested according to NACE TM0198 and ASTM G129 [44,53]. The deterioration by SCC is fundamentally evaluated by differences in the behavior of the mechanical properties of the specimen in tests conducted in a particular environment (soils or solutions from external and internal environments, respectively) from that obtained from evaluations obtained in air, including equations and criteria used in the assessment [44,64,65]. The tendencies (ratios) of metals to ductility parameters such as plastic strain-to-failure ratio = (plastic strain-to-failure in the test environment/plastic strain-to-failure in air) × 100 must be addressed [44]. Thus, ratios of plastic strain-to-failure about 0.8–1.0 means low susceptibility to EAC, whereas low values (i.e., <0.5) show high susceptibility. To increase the SCC resistance, values of plastic strain-to-failure ratios near 1.0 is recommended. Spectroscopic and mechanical evaluations of the specimen are recommended to establish whether or not SCC must be achieved from a plastic strain-to-failure ratios lower than 0.8. Thus, crack properties must be tested on the longitudinal section of the gage.

In order to achieve a homogenization related to testing techniques in industrial and academic fields, a status of standardization was reported by Kane and Wilhelm since 1993 [66]. Recently, the techniques widely used for assessing the SCC susceptibility of carbon steels are the SSRT and the constant load testing (CL) [44,53,67]. In the first, a constant extension rate tests (CERT) in an Inter-Corr machine type M-CERT with a specific load capacity (44 kN), total extension of 50 mm, and a strain rate of 1 × 10^−6^ s^−1^ for carbon steels was used and a SCC analysis may be carried up to 3 days of exposure time [44,53]. Typically, cylindrical or rectangular SSRT tensile specimens may be machined for assessing SCC phenomenon depending on the tension test machine and experimental conditions. The specific measures of the SSRT test specimen (size, shape, and type), experimental procedure, and corrosive media detail are given in NACE Standard MR0103, NACE Standard TM0169, NACE Standard TM0177, ASTM G129, and NACE Standard TM0198 [61,62,63,64,65,66,67,68,69]. In the second, bent beam, C-ring, U-bend, specimens for weldments, precracked wedge open loading type, precracked, cantilever beam, and Bent-Beam are some recommended specimens for assessing SCC of carbon steels [61,70,71,72,73]. In this way, a constant load (tensile stresses) is applied on specimens exposed to solution promoting cracking for an evaluation of long duration (up to one month). 

Other SCC testing detail is given in NACE Standard TM0284 [74]. New materials (including polymer compounds or composite coatings), good test duplication including the reference (inert environment such as air), allows SCC susceptibility of materials to be analyzed at relatively short test-time, uses terms familiar to engineers or researchers, enables corrosion testing of weldments specimens to be analyzed, and failure analysis by cracking in operating pipelines ruptures must be determined. They can be tested in the absence (reference) and in the presence of corrosive solution. In addition, combining electrochemical techniques can be applied to the assessment of simultaneous corrosion process and tension tests [8]. However, it is extremely important to isolate the electric contact of the specimen from all tension test machine structures. This enables different cracks, brittle zones, fractures, and deformed grain zones to be studied by electrochemical and spectroscopic methods, which are more reliable in comparison with those in CL. In addition, some cautions and limitations are: (1) anomalous behavior of the SCC test is related to inadequate control or measurement of strain rate or potential, (2) not correlation by combining electrochemical (impedance or polarization curves) and SCC methods is given for interpretation data from operating pipelines (field scenarios and dig data in buried pipeline such as longitudinal stress, circumferential stress, axial crack, transverse crack, circumferential crack, oblique, and not oblique-circumferential neither oblique or circumferential stacking of axial cracks), (3) usually applicable to high strength materials, (4) may oversimplify initiation stage of cracking as well as specimens could be large and expensive to machine compared with some smooth specimens [1,75].

In addition, Afanasyev et al. [45] proposed a cyclic test simulating a real load from a field scenario to determine the variables affecting the cracks and fractures (in the absence of corrosive environment) concerning the wide variation of lengths and widths [73]. In this test, the methodology consists of cyclic tests of several damaged samples from real field conditions were four-point loading conditions is applied [73]. The applied load (cyclic pressure change or cyclic test modes) can be simulated from the whole range of pressure fluctuations from the upper and lower half-periods over a representative period of a specific operating pipeline.

### 2.4. Complementary Test Methods for Assessing SCC

Some experimental tests related to SCC and electrochemical responses have been reported [68,76,77,78,79,80]. Most notably, standard recommended practices for complementary external SCC studies may be ASTM G57 and ASTM G187 for the soil resistivity determination by using a soil resistivity meter; moisture content using a gravimetric method which is recommended by ASTM D4959, and ASTM G4643 using direct heating or microwave oven heating, pH value using a potentiometer and the ASTM G51, redox potential analysis using a potentiometer and ASTM G200, texture characteristics, oxygen concentration, metal and anions content using the inductively coupled plasma (ICP) and ion chromatography, cathodic disbonding evaluations recommended by ASTM G8, ASTM G 42, ASTM G80 or ASTM G95, as well as the NACE Standard TM0497 related to criteria for cathodic protection on buried pipelines, ASTM G62 for holiday observation in coatings, NACE Standard TM0106 for studies of microbiologically influenced corrosion, and ANSI/NACE Standard RP0502 for pipeline external direct assessment [81,82,83,84,85,86,87,88,89,90,91,92,93,94,95]. In addition, the electrochemical data acquired from the cyclic tests proposed by Afanasyev et al. [45] were studied using fractography analysis from cross section samples. Based on the spectroscopic analysis this is a technique used to obtain elemental composition of corrosion products (oxides) from maps of elements distribution in a SCC [45]. The standards recommended practices for complementary evaluation of I-SCC are: NACE Standard SP0206 for direct assessment of internal corrosion in pipelines transporting dry natural gas, and NACE Standard SP110 for direct assessment for pipelines in wet natural gas [96,97]. Additional standard recommended tests for internal studies include: texture characteristics, oxygen concentration, metal and anions content using the inductively coupled plasma (ICP) and ion chromatography, redox potential analysis using a potentiometer and ASTM Standard G200, ASTM Standard D4294 for petroleum and petroleum products containing sulfur compounds, ASTM D287 or ASTM D1298 for American Petroleum Institute (API) gravity test in hydrocarbons to know its viscosity grade, ASTM D664 for acid number of hydrocarbons, oil viscosity, oil density, ASTM D5002 for density, relative density, and API gravity of crude oils, water (water cut) and sediments by centrifuge method, pH of water, water in crude oil by distillation, metals, chlorides, sulfates, conductivity and resistivity in water, iron in water, bacteria, biocides, among others [86,87,98,99,100,101,102,103,104,105,106,107,108,109,110,111,112,113,114,115,116].

### 2.5. How to Prevent SCC and Factors that Affect Its Evolution

Pipeline companies, industrial groups, and researchers have observed ways to prevent, detect or remove SCC before they grow to the point where pipeline failure can occur. Perhaps, the pioneer and most advanced country in these studies is Canada thorough the National Energy Board, followed by USA [54]. This has resulted in the publication by the Canadian Energy Pipeline Association of a set of practical recommendations and more recently of a comprehensive review of SCC that has been carried out by Michael Baker and associates on behalf of the U. S. Office Pipeline Safety [117]. The results of research, techniques used, and experience gained have resulted in a continuous increase in useful information. This will continue to happen this way as long as there is additional research in this area. All these points together have improved the understanding of SCC and have helped to formulate procedures for the study of SCC mechanism. In order to reduce the risks due to SCC, you should do:A suitable selection of the pipeline steel;Stress control (residual, operation, external loads);Remove critical species from the environments (H_2_S, CO_2_, Cl^−^, among others);Suitable cathodic protection ranges (avoid over cathodic protection);Select the appropriated coating.

Figure 2 describes the main factors that must be considered to prevent that internal and external SCC develop in a pipeline. The parameters that must be considered to prevent SCC development in pipelines according to factors mentioned above are describe in the following section.

#### 2.5.1. Suitable Material Selection

The material selection is one of the main factors that must be considered to prevent SCC development in pipelines, and these factors must be taken into account:Use materials with low SCC susceptibility;Good quality (low inclusions, porosity, among others);Proper toughness steels;Suitable microstructure to avoid SCC.

Taking into account the last studies considering the SCC susceptibility of the material, we have the following remarks:The greater the API steel grade, the higher strength but greater the susceptibility to SCC;All low carbon steels have certain degree of SCC susceptibility;The greater the hardness of steel, the greater the susceptibility to SCC;Higher quality steels have lower susceptibility to SCC;Steels with microstructure of bainite and acicular ferrite have a lower susceptibility to SCC.

#### 2.5.2. Tension Stress Control

The tension stress in pipelines must be controlled and reduced in order to avoid development and growth of cracks in materials susceptible to SCC. The parameters that could be considered are:Proper heat treatment to reduce residual stresses;Cold expansion to reduce residual stresses;Pressure fluctuation control;Reduce operating pressure.

#### 2.5.3. External and Internal Environment

The internal and external parameters related to the environment must be considered in order to reduce the possibilities of nucleation and growth of cracks in materials susceptible to SCC.

The next are some of the parameters that must be addressed:Remove corrosive species in the fluid transported like Cl^−^, H_2_S, CO_2_, Sulfates.Remove standing water in pipelines;Soils with high content of carbonates (CO_3_^2−^) are prone to develop SCC;Use inhibitors sometimes helps to avoid corrosion and SCC;Proper coatings in corrosive soils.

#### 2.5.4. Proper Cathodic Protection

The proper levels of cathodic protection are important to avoid dissociation of water that generates monoatomic hydrogen that ingresses in the material producing embrittlement. The parameters that must be considered are:Suitable cathodic protection;Avoid over cathodic protection;Resistant coating for over cathodic protection;Interferences of direct current.

#### 2.5.5. Suitable Coating Selection

The suitable selection of coating is one of the most important parameters that must be taken into account to prevent SCC development in pipelines, because the coating is the physical barrier that must be avoid the contact between the electrolyte and metal materials, and these factors must be taken into account:Use high performance coatings;Good surface preparation;Coating with good adhesion;Use of compatible coatings.

### 2.6. How SCC May Be Controlled in Carbon Steel Pipelines?

In field practices, the external corrosion for buried, submarine, and atmospheric pipelines may be controlled using anticorrosive and improved coating (enhancement polymer or epoxy materials) as a physical barrier, cathodic protection systems, materials, and environment.

#### 2.6.1. Coatings

To protect the external damage of carbon steels, organic coatings continue to be a topic of great research interest. Organic and epoxy resin coatings have low shrinkage and deterioration, and high chemical and mechanical characteristics. They can be applied on ships, oil tanks, seabed and buried pipelines [118,119]. The purpose of the coating is to prevent the corrosion and avoid the direct contact between the carbon steel surface and the external variables containing hydrogen to promoting SCC [1,4,5,17,24]. The type of coating that is applied commonly on external surfaces of the operating pipelines are epoxy, wax, polyolefin resin coating, coal, cement, composites, or alkyd, such as fusion bonded epoxy (FBE), liquid epoxy, urethane, asphalt, coal tar, multilayer or composite coating, poly(methyl methacrylate) PMMA, asphaltic compounds, polyethylene (PE), viscoelastic (plasticity of coating used) polymer, among other [1,3,5,7,9,10,120,121,122,123,124]. They are applied to protect the external pipeline with good mechanical properties, poor permeability to water, effective electrical insulation, good adhesion, adequate thickness, and high resistance to degradation with time [1]. Based on FBE this is an epoxy coating used to resist the mechanical damage during the handling, storage, construction, and installation in new large diameter pipelines. It is applied to maintain high resistance to hydrocarbons, acids, and alkalis [3]. However, FBE is susceptible to cathodic protection, producing blistering or shielding effects [1]. Deteriorated or aged coating could occur when external SCC is an influence, because the performance of the coating may be affected by the residual or applied stress of the operating pipelines, and this depends on the exposed soil or atmospheric environment type, including the environmental changes, protection systems, possible incorrect mixing ratio of the coating, and incorrect surface degreasing and preparation.

#### 2.6.2. Cathodic Protection

One way of reinforcing and supporting the external coatings against the external damage is the application of an impressed current or sacrificial anodes (cathodic protection systems). In this way, the external corrosion for coated and buried pipelines is controlled by the type of the coating, soil, seabed, and its physicochemical modification due to change of season, which in turn localizes the damage condition of the coating as well as the cathodic protection, that could be displaced toward negative behavior. The negative behavior is related to the presence of carbonates, disbonded coating, hydrogen production, external SCC, blistering and shielding effects on the high performance coating (Cheng, 2013; Beavers and Thompson, 2006). The standard used to achieve the external corrosion control on buried pipelines is NACE SP0169 [125]. Some Standard methods for disbonded coating studies are ASTM G8, ASTM G42, ASTM G 80, and ASTM G95 [88,89,90,91].

#### 2.6.3. Materials and Microstructure

Single-phase microstructures (excluding precipitates) generally provide a higher resistance to SSC than a multiphasic structure. Most of the fractures produced by the SSC often begin in internal locations of the material, in places of high concentration of internal stresses, while the SCC frequently starts on the surface of the metal from cracks or pitting. In the aspect of hydrogen embrittlement, the effect of damage occurs in two places, at the tip of the crack directly generating the material’s cohesion, or internally between the sites of hydrogen run-off given mainly at the grain edges and defected crystalline. The exact composition of the alloy and its microstructure, as well as the heat treatment applied, have a marked effect on the development of the SCC. The process of embrittlement by hydrogen on steels is somewhat significant to the strength of the metal, additionally, these materials have a lower critical stress intensity factor, with the fracture occurring more quickly being associated with a smaller crack size. The type of alloy elements in carbon steel pipelines is especially important to the SCC mechanism, as over time these changes can occur due to the natural aging of the materials. An effect such as the formation of segregated phases on the grain edges of the elements of alloy can be distinguished more clearly in the span of about 20 years of the pipelines in service. An example of this is greater occurrence of fractures and leaks associated with the SCC that have occurred in the United States of America and Canada [126]. The microstructure of the carbon steel pipelines contain a ferrite-pearlite phase. Homogeneous microstructures may be less susceptible to SCC [127,128,129,130,131,132]. However, these steels have traces of sulfur forming manganese sulfide inclusions (MnS). Other inclusions are aluminum oxides, calcium oxides, and calcium sulfide. According to Liu et al. [133] the origins of the crack initiation sites must be due to the inclusions. The authors said that inclusions enriched in Al_2_O_3_ and SiO_2_ can form a hard, brittle, and incoherent character on metal matrix. Micro cracks and interstices are fundamentally formed in the grain boundary, and around of the inclusions. Asahi et al. [128] measured the threshold stress of X52, X65, X80 steels by different thermal treatment. The authors concluded that the thermomechanically controlled processed (TMCP), and quenched-tempered (QT) on these steels, achieved more uniform microstructures, were less susceptible to SCC. Thus, the microstructure of carbon steel pipelines has been considered one important factor affecting SCC initiation and propagation [64,65,127,128,129,130,131,132]. Therefore, the SCC resistance of carbon steel pipelines can be increased by eliminating the inclusions or precipitates content to avoid the nucleation sites for corrosion pits on carbon steels [64,65,132,133,134].

#### 2.6.4. Environment

SCC is attributed to the soil and the fluid transported—the external environment and internal environments—which concern different external and internal corrosion mechanisms. SCC is highly relevant to economics, environmental damages, pollution and safety needs, which are all expected to be monitored globally [54,65,126,135,136,137,138,139,140,141,142,143,144,145,146,147,148,149,150,151,152,153,154,155,156,157,158,159,160]. Thus, the principal problem induced by ions—mainly, naphthenic acid content, partial pressure, temperature, CO_2_ and H_2_S—is the E-SCC and I-SCC.

Concerning the environments relating to physicochemical properties, nearly neutral pH and high pH in soils and representative soil solutions containing carbonates are some conditions where SCC occurrence were highlighted [159]. SCC increases with the high cathodic polarization simulating an overprotection on carbon steel [14,161,162,163]. In this case, a good coating should be applied. The nearly neutral pH SCC of carbon steel pipelines is related to tape disbonds in coatings, which are able to produce carbonates from CO_2_ [4,6,7,17,19,124,161,163,164]. High pH values are related to SCC failures where CO_3_^−2^, HCO_3_^−^, and alkaline SCC damage is conferred [21,27,165]. In addition, Yin et al. [166] mentioned that CO_2_ gas can dissolve in the presence of water producing carbonic acid, which is an aggressive solution to carbon steel. Hydrogen production on carbon steels from sulphate-reducing bacteria or acidic soil solution sources may occur [15,18,24,133]. Thus, possible SCC occurrence under the defected coating from pipelines could be achieved promoting an extreme condition [167]. This is because damaged coating is a local defect where the cathodic protection level (low or high) is strongly influenced. This negative manifestation is increased when the external applied coating on pipelines is corrugated in specific places during the construction and installation of the pipelines in the form of mechanical damage as was reported by Afanasyev et al. [45]. When the physicochemical properties of the soils (including air, oxygen, water, and solid particles), and their possible mechanical effects become random and prolonged, the corrugated coating are subject to an abrupt dried or saturated conditions (depending on the seasonal fluctuations in a specific location) resulting in more corrosive environment as extreme situations [9,11]. Carbon steel corroded surfaces can be attributed to the suddenly changes of the environment such as seasonal period, which in turns modifies the levels of the cathodic protection systems and the physicochemical properties of the soils [9]. The above means that buried pipelines can be affected by extreme stages during the change of season. Soils containing variable moisture (low and high weight percent of water) can modify their physicochemical properties for each season of the year and they could provoke a random climate for buried pipeline steel. For that reason, random modifications of the mechanical properties, ion concentrations of the soil, and electrochemical responses on pipeline steel-soil interface are expected. For geological purposes and scientific studies, it is important to mention that natural soils should be collected at the bottom of a buried pipeline (around 1.2 m in depth) from a right of way (ROW) where severe external corrosion damage could be identified [10]. Romanoff [168] has mentioned that the external corrosion for metallic structures is affected by metal types in the absence and in the presence of coating immersed in different soils located through the United States. In addition, Cole and Marney [25] reported that the corrosion process for metallic materials exposed to soils can be attributed to electrochemical reactions, oxide effects, the impact of the physicochemical properties such as temperature, moisture content, salinity, pH value, and void fraction, mainly. The authors concluded that among all parameters, different codes, models and methodologies occupies important tools to predict and prevent the external corrosion for buried pipelines transporting water, sewerage and oil and gas systems.

In field scenarios, the internal SCC of carbon steel pipelines could be arrested or minimized using a field separator device to remove the corrosive gas, hydrocarbon, water cut (oilfield produced water containing high salt contents), and possible sludge containing solids, metals, and sediments, which include a desalting and dehydration processes. Internal SCC occurrences could be activated by the nature of the fluid (fluid composition) containing CO_2_, H_2_S, organic acids, water content, density (gas and liquid), viscosity, dissolved oxygen, microorganism, suspended solids, and field sludge [169]. Moreover, the stronger interaction with the pipeline operating condition including the temperature (inner wall, fluid, gas, and liquid), the rate (gas, liquid, deposition, erosion velocity), heat transfer, surface shear stress, flow pattern (turbulence intensity), and pressure is highlighted. In the last variable, fouling propensity influence is attributed to the suspended solids, and field sludge resulting in an over-operating pressure during the production, gathering, storage and transportation of hydrocarbons through pipelines. As a result, an increase in operational cost and product yield reduction is conferred. Thus, anticorrosion measures using chemical treatments (corrosion inhibitor programs) and direct (corrosion coupons, failure frequency, inline inspection using smart pigs, radiographic testing, ultrasonic testing, among other methods) or indirect (water cut, liquid analysis, microbiological components, inhibitor traces, gas analysis, solids analysis, hydrogen flux, temperature, pressure, among other data) corrosion monitoring for internal pipelines is required [94,116,170,171,172]. It is well known that an activity/operation related to fouling tendency in oil pipelines is the in-line cleaning systems using a pig tool (pigging for cleaning the pipeline), same with gas pipelines but uses a sphere not a pig [46].

### 2.7. SCC Originates from Pits

SCC testing is related to pit-crack transition processes including microvoids and other types of discontinuities, as is shown in Figure 3 [59]. Thus, unpredictable pits-cracks initiation and propagation including the plastic deformation, tensile residual stresses (stress intensity parameter), and corrosion fatigue and the stage of SCC models are taken into consideration [59,173,174,175,176,177,178,179,180,181,182].

## 3. Parameters Reported in the Literature Regarding the External SCC

In empirical and theoretical practices, corrosion damage modeling and the effect of several field scenarios are related to SCC analysis, where the operating pipelines exposed to different external variables (types of electrolytes, temperature, pH, among other data) have been studied and reported in the literature [20,168,183,184,185,186,187,188]. Nowadays, SCC of carbon steel pipelines is increasing in popularity, and the original mechanical properties of the steels and their initial electrochemical interactions with natural soils, representative soil solutions, or any solution promoting cracking have been described including SCC, intergranular-SCC (IG-SCC), transgranular-SCC (TG-SCC), SSC, HIC, hydrogen embrittlement from testing and characterization of carbon steels in hydrogen or sour testing [8,9,10,11,189]. In this way, metallurgy, physicochemical properties, codes, mathematical models, and electrochemical responses to predict external corrosion in terms of pH value, redox potential, resistivity, and pipe-soil potential can be considered [184,185,186,190]. Some empirical (generic) crack assessment models regarding SCC are API 579 and BS 7910 [191,192]. However, SCC occurrence is still not well understood (including the generic SCC models) and requires more experimental research. Carbon steel pipelines exposed to aggressive and external variables are the main causes of unpredictable cracking and fracture initiation mechanisms. Additional information on physicochemical, electrical, and mechanical testing and evaluations for external environments of operating pipelines may be found in Section 2.3. Table 1 shows some external parameters and environment to promoting the external cracking [43,193,194,195,196,197].

According to this Table 1, the external corrosion for buried pipelines is attributed to different types of physical, chemical, mechanical, electrical and biological environments. As a result, the possible reduction of the original mechanical features for pipeline steels could occur [8]. After a brief review, the experimental data based on the range of variables studied and reported in the literature regarding the external SCC of carbon steel pipelines were selected. Data acquired from the reviewed experimental part are documented in Table 2, Table 3, Table 4, Table 5 and Table 6. Table 2 and Table 3 are related to reviewed physicochemical properties [4,5,6,7,9,11,14,15,16,17,18,19,59,124,125,126,127,128,129,130,131,132,133,134,135,136,137,138,139,140,141,142,143,144,145,146,147,148,149,150,151,152,153,154,155,156,157,158,159,160,161,162,163,164,165,166,167,168,169,170,171,172,173,174,175,176,177,178,179,180,181,182,183,184,185,186,187,188,189,190,191,192,193,194,195,196,197,198,199,200,201,202,203]. Variables listed in Table 4 are related to reviewed mechanical properties, while Table 5 and Table 6 are associated to reviewed electrochemical properties [4,9,11,15,16,17,18,19,23,24,36,124,133,159,163,164,165,198,199,200,201,202,203].

According to Table 2 and Table 3, some coating types such as FBE, PMMA, PE, and viscoelastic polymer in the absence and in the presence of damage were used to study the external corrosion process [4,5,6,7,9,124]. Simulated soil solutions containing carbonate compounds were more documented than the natural soil. The reviewed pH values were about acidic, neutral, and alkaline media (4.0 to 9.6). Moisture content values were around 1.5 to 52.3 wt % for some soil types, while the redox potentials were about 50 to 373 mV vs. Ag/AgCl electrode.

Regarding the mechanical properties, Table 4 showed the YS, UTS, elongation (E), and elongation plastic (EL) which are related to SCC. They may be determined using NACE Standard TM0198 and ASTM Standard G129, respectively [44,53]. In some cases, different heat treatments for X70 steel were studied, the elongated alternating grains of pearlite and ferrite phases and the presence of inclusions were reported elsewhere, microcracks and energy dispersive spectroscopy (EDS) analysis were analyzed, and micrograph for fracture and calculation of crack and toughness propagation rate were documented [18,19,21,22]. In the case of mechanical tests, a strain speed (v) of 1 × 10^−6^ s^−1^ is accepted as typically strain rate for SCC characterization by using slow strain rate tests [8,44,53]. Table 5 and Table 6 show that poor corrosion rate (CR) and charge transfer resistance (R_ct_) data were obtained. However, CR values were around 0.001 to 4.64 mm/year depending on the electrolyte. As regards the brief experimental summary, exposure time, scanning electron microscopy (SEM) examination and hardness Vickers (HV) for X60 (201 HV) and X65 (221 to 288 HV) steel exposed to synthetic soil solution were reviewed [8,160]. A uniaxial elastic stress study of X100 steel immersed in a near pH medium was published elsewhere [198]. This study concluded that not fracture during loading where the critical failure strain around 0.00357–0.00417 was reached. In addition, a special cell arrangement for electrochemical studies and SSRT was reported earlier [199]. Thus, a natural soil containing corrosive ions, organic content (22,600 mg/kg soil), whole nitrogen content (910 mg/kg soil) and total salt content (464 mg/kg soil) were analyzed [199]. It revealed that the stresses and the presence of sulfate-reducing bacteria (SRB) enhanced the SCC susceptibility on steel surface. Wu et al. [200] reported the mechano-chemical study for pipeline steel immersed in a special solution showing a microbiological corrosion by SRB related to the *Desulfovibrio desulfuricans*. The authors concluded that different micro-voids-cracks should be preconditioned in certain stages from the mechanic-chemical study (tensile load applications). It was published earlier [200] that the interaction between SRB behavior and strain growth rate could not be correlated. A microbiological corrosion study for pipeline steel exposed to soil and under yield stress was reported in a previous work [99]. In this case, SRB and culture solution with pH around 7.0 to 7.2 were selected. Fu and Cheng [4] reported different test solutions containing different carbonate-bicarbonate concentrations with pH and conductivity around 9.4–9.6, and 21.8–61.5 mS/cm, respectively, in order to tests the electrochemical and SCC behavior for X70 steel with a disbonded coating. The authors concluded that the steel could be passivated, and an active-passive transition may occur. Additionally, these authors reported that the corrosion potential of the steel was shifted toward negative values, and the current density increased using the cathodic protection test, and the stress enhanced as well. In some prepared solution such as NS4 near-neutral pH value containing bicarbonates, chloride compounds and sulfate has been reported [201]. In this study, corrosion under deposits affected by cathodic and anodic reactions for X70 steel was analyzed. Experimental data revealed that the deposit formed on steel surface was porous and non-compact in nature inhibiting the absorption and permeation of hydrogen. SCC could start from the pits and dissolution sites that must be formed underneath the oxides films. However, the deposit increased the corrosion effect. Other work [202], carbonate-bicarbonate media, various solutions were proposed to SCC characterizations: 0.5 M Na_2_CO_3_ + 1 M NaHCO_3_ pH = 9.32. Electrochemical results suggested that in the absence of alternating current cracks propagation was intergranular and the SCC mechanism was attributed to the anodic dissolution [202]. While, trans-granular cracks were obtained in the presence of alternating current. These mechanisms were attributed to the alkaline pH value, dissolution and hydrogen embrittlement behavior. In another study, sodium-carbonate solutions was used to assess the corrosion process for a stressed section cut from U-bend X70 steel specimen [162]. It concluded that pitting effects and cracks initiation occurred by the deformation-induced stress in alkaline medium. Carbonate compounds had an inhibitive behavior at low deformation-induced stress blocking the stressed zones. In addition, tensile stress increased the metal dissolution more considerably than compressive stress, resulting in more calcareous compounds.

The original mechanical character of the steel is an important parameter in the SCC mechanism. Thus, the original properties could be diminished when pipeline steels are exposed to corrosive environments, and a stronger interaction with the electrolyte type and its concentration, pH value, hydrodynamic condition, temperature, and gas pressure) may occur. Thus, the external corrosion damage initiation may be expected on susceptible anodic sites [9,11]. That is because different damage of the microstructure, and electrochemical responses of the cracked tip exposed to natural soil or simulated soil solution can be obtained [8,160]. On the other hand, external corrosion studies were developed using high anodic polarization or microbiological effects [162,164,200,202]. In a work, different working electrodes of a ruptured pipeline during 19 years of being buried in near-neutral-pH solution (since 1980 up 1999) were used [161]. Moreover, different external crack depth and crack width for external SCC on carbon steels were reviewed. Microcracks, secondary cracks, and crack growth must be expected as well [160,164,172,198,199,200,201], crack initiation [162], sharp intergranular cracks and anodic dissolution crack [159], cracks initiation [165], shallow cracks [124], intergranular cracks at elevated temperatures [32], intergranular, and trans-granular cracks narrow and shallow cracks [14,23,199,202], “linked” and parallel cracks [161] for external corrosion were reviewed.

The reviewed electrochemical tests included three-electrode set up and different geometries of the working electrode such as flat tensile specimens, sheet specimens, rod specimens, U-bend steel specimen. Preliminary conclusion from external SCC suggested that microstructure of the carbon steels, coating type, pH value, redox potential, and moisture content simulating the stages of season of the year, electrolyte type from each soil type are a really important parameters affecting the cathodic potential variations and the possible hydrogen evolution in underground pipelines under dry and rainy conditions. Some studies related to the integrity approach (stochastic and Bayesian models) by semi empirical or theoretical modeling for low carbon steels exposed to soil have been documented [183,184,185,186,190]. However, no external SCC model was reported. Additionally, all the above did not include the social problems where corrosion failures in pipelines are attributed to illegal tapping.

In field scenarios from buried pipelines in México, and in accordance with the standard recommended practices for complementary external SCC studies, some determined parameters include: moisture values up to 50 wt %, pH around 4.0 to 9.0, mass loss up to 20%, exposure time up to 40 years, coated pipelines (up to 36 years), range of temperature about 65 °C, ultimate tensile strength (UTS around 500 to 900 MPa), yielding strength (YS around 396 to 600 MPa), potential around −850 mV vs. Cu/CuSO_4_ for buried pipelines, 950 mV vs. Ag/AgCl for seabed (submarine) pipelines, and redox potential about 600 mV vs. Ag/AgCl could be achieved in pipeline exposed to natural soils where seasonal fluctuations is expected [11,92]. Concerning biological parameters (biocorrosion), the microorganisms affecting the SCC phenomena are related to SRB, sulfur-sulfide oxidizing bacteria (SOB), iron-oxidizing bacteria (IOB), iron reducing bacteria (IRB), organic acid-producing bacteria, and acid-producing fungi as was reported by Cheng [1]. In the oil field, the term bacteria is related to a biofilm formation causing negative effects [204]. Concerning SRB this is an anaerobic microorganism which is able to reduce sulphates toward sulfides producing hydrogen sulfide (H_2_S) at 25 to 50 °C, and it is commonly found in internal pipelines up to 60 °C and high salinity (produced water, connate water or brine water) [205,206]. SOB and IOB are aerobic microorganisms. IRB can produce Fe^3+^ corrosion products (rust) and is able to form anaerobic zones to favor the growth of biofilm by SRB when both bacteria are interacted [34,205,206]. The corrosion occurs along the localized sites of the metal where induced product from the adsorption, growth, division, metabolic waste, and decomposition is caused by the bacteria to conduct other forms of corrosion such as corrosion under deposit, pitting, crevice, galvanic, and intergranular deterioration [34].

## 4. Parameters Reported in the Literature Regarding Internal SCC

Table 7 shows some internal environments affecting operating pipelines to develop SCC. Clearly, the environments (physicochemical, mechanical, electrical, and biological) of operating pipelines, include the quality of the hydrocarbon such as API gravity of hydrocarbons, oil viscosity, oil density, pH of the oilfield produced water (water cut), types of electrolyte into the oilfield produced water, as well as the operating pipeline conditions, mechanical behavior attributed to tribology (friction, wear, lubrication, and erosion), electrical, and biological parameters.

Moreover, a review of the physicochemical, mechanical, and electrochemical variables studied and reported earlier are related to the I-SCC of carbon steel pipelines were carried out. The experimental part based on the physicochemical, mechanical, and electrochemical parameters affecting the internal corrosion were listed in Table 8, Table 9 and Table 10 [37,38,208,209,210,211,212,213,214,215,216].

Table 8 shows the physicochemical features of representative solution simulating electrolytes commonly found in internal pipelines. The reviewed pH values were around 3.4 to 11.9 and depending on the electrolyte type. As regards mechanical properties, Table 9 shows that the yield stresses were about 431 to 690 MPa, and the ultimate tensile strength of 480 to 1050 MPa were obtained. Table 10 showed that CR was around 0.01 to 98.5 for different metals, and the corrosion potential (E_corr_) was around −600 to −864 mV referred to saturated calomel electrode (SCE). While poor R_ct_ values were obtained. Additionally, corrosion inhibition of X52 steel exposed to sulfuric acid containing different concentration of four corrosion inhibitors was studied [208]. In another study, CR values obtained for X80 steel exposed to sodium sulphate containing CO_2_ gas (Table 10) were calculated by the corrosion current density reported in a previous work [37]. CR values obtained for X80 steel exposed to NaOH at 100 °C (Table 10) were calculated by the corrosion current density reported elsewhere [209]. In another study, CR values obtained for X60 steel and Inconel 625 alloy both exposed to NACE test solution were determined by using weight loss tests [38]. E_corr_ values obtained for X65 exposed to 3.5 wt % NaCl containing different H_2_S-CO_2_ gas ratio were around −670 to −590 mV referred to Ag/AgCl reference electrode [215]. An investigation related to heat affected zone (HAZ) for X70 steel different base metal and weld metal was reported [210]. In another study, four API steel grades were tested using a recommended solution from NACE Standard TM0284-2016 during 96 h prior to constant elongation rate tests with a strain rate of 0.003 mm/min [211]. Thus, susceptibility to SSC and HIC were studied. Other test solutions used for the constant elongation rate test may be prepared according to NACE Standard medium.

The inhibition of X52 steel exposed to 1 M H_2_SO_4_ was investigated by Morales et al. [208]. In this work, an optimal corrosion inhibitor concentration, and a maximum inhibiting efficiency were obtained. In another work, the heating–cooling cycles on simulated weld heat affected zones for X80 steel exposed to chloride, bicarbonate, and CO_2_ solutions at different pH were reported [37]. Moreover, SCC of various carbon alloys exposed to alkaline (8.5 M NaOH at pH 13.2 and 100 °C) solution was studied, too [209]. This investigation concluded that susceptibility to caustic SCC was attributed to carbon element, and oxidation effects; but small beneficial effects was obtained when titanium was added into grain boundaries inhibiting the formation of magnetite (Fe_3_O_4_). A representative test solution was prepared in order to study the trans-granular SCC on X70 steel exposed to carbon dioxide-containing ground water type A [212]. The solution was used to study the CO_2_ to HCO_3_^−^ and/or CO_3_^2−^ effect attributed to pH changes [212]. Regarding the electrolyte used to re-passivation tests of X65 steel exposed to fuel grade ethanol, a specific solution was prepared from three baseline test environment [213]. One of them contained 1.0 vol% water, 0.5 wt % methane, 56 mg/L acid acetic and balance with 200 proof ethanol reaching pH = 5.6. The others contained different amount of chloride in part per billion (ppb) reaching pH around 7.6–7.7. Their calculated resistivity values were around 0.5–40 kΩ-cm, and they depended on its type and concentration fuel grade ethanol. In another work, the crack advance for X70 steel based on stress corrosion cracking and fatigue was analyzed using the test rig clevises [214]. X70 steel was placed in an autoclave with 4.5 L of total fluid at 30 °C. Different cycles were used to simulate the stress cycles for detecting crack growth, and an initial test for X70 steel was carried out using maximum stress intensity factor (K) and 20 Hz [214]. To evaluate the electrochemical behavior for X65 steel exposed to 3.5 wt % NaCl containing different H_2_S-CO_2_ ratio, Pessu et al. [215] used Tafel plots. It concluded that different mechanisms of H_2_S-CO_2_ corrosion were affected by temperature. It is important to point out that different crack depth and crack width for internal SCC on carbon steels were reviewed. Thus, secondary cleavage cracks, nucleated cracks, intergranular cracks, stress corrosion cracks, longitudinally orientated cracks and shallow cracks, circumferential cracks, stress corrosion cracks, deepest cracks, orientated cracks, interdendritic and secondary cracks, shallow cracks, yielding stress corrosion, small and large cracks, yielding stress corrosion cracks, and stress corrosion cracks, propagated cracks, secondary cracks, cracks propagated trans-granularly were studied and reported in the literature [38,39,208,211,212,213,215,216,217,218,219,220,221]. Moreover, mechanical properties for carbon steels can be affected by the hydrogen presence causing cracks on external pipeline surfaces by HIC [159,222]. The reviewed internal corrosion data showed that there are some models to predict the corrosion damage and determining of the remaining strength of corroded surfaces in an operating pipeline according to values of parameters such as operating pressure and temperature, gas flow, and the corrosiveness of the electrolyte; however, internal SCC is not considered. Erosion, generalized, and localized (pitting) corrosion are only allowed to the empirical and theoretical internal corrosion models. However, why is in-line inspection (ILI) technology an important tool in operating pipelines? Because ILI (magnetic flux leakage, pipeline pig) technology is a project objective for identifying and characterizing pipelines features such as anomalies, indications, defects, or integrity threats [97,172,223,224]. Thus, development of protocols from data integration and evaluation is highlighted. Then, a report of the integrity assessment scenarios (crack modeling, metal loss, and technical recommendation to carry out the inspection from failure analysis diagram) and prioritization (features and selected sites to excavate and repair) is conferred [191]. However, hydrostatic pressure testing (proof of service), long-range ultrasonic testing (LRUT), guided wave technology (GWT), automated ultrasonic testing (AUT), and manual ultrasonic testing (UT) are other technologies to support the evaluation of ILI technology given a complete scenario of the damaged or corroded surfaces of the pipelines [97,223]. Internal corrosion models are commonly applied to analyze the parabolic profile of mechanical and corrosion damage, regression analysis, holistic analysis, and mechanistic models based on remaining life of pipeline systems, among other data. Internal corrosion is attributed to pressure and temperature concerning the flowrate in thousands of barrels per day (TBD). However, internal pipe wall thickness is commonly affected by different SCC mechanisms. Therefore, hydrogen and CO_2_ (pH < 7.0) are two main electrolytes promoting internal SCC [27,215].

After a brief review, it is possible to note that the internal SCC phenomena could be affected by the hydrogen source that comes from sour media or brine sour media (hydrogen sulfide, and carbon dioxide), hydrochloric acid, sulfuric acid, sodium hydroxide, organic acids, internal bacteria induced corrosion, cathodic polarization, and by the over-operating pressure, and elevated temperature. Internal corrosion damage for pipeline steels has been analyzed using several corrosion models, numerical data, different mathematical modeling strategies such as hypotheses models and theories [169,223,225,226], historical data and internal corrosion inspection data of the pipeline [96,97], acquired code from artificial neural networks which are used for a non-lineal corrosion rate prediction model as well as to weigh and refine the influencing factors using Grey relational analysis methods [169]. Additionally, top-of-line corrosion models [169], stochastic models, power or exponent law (empirical or semi-empirical models), Bayesian analysis, statistical models and mechanistic models or electrochemical reactions [184,227,228,229,230,231,232] have been used. In field practices, the operating pipelines are exposed to different internal variables, which in turn means they react in many ways, as shown in Table 7. Thus, internal SCC could be attributed to field sludge (fouling tendency influence) plugging the transport process of the hydrocarbons and increasing the pressure gradient (pressure drop). This problem is increased when an incorrect operating pressure (overpressure) may occur and more flow in thousands of barrels per day (TBD) of the hydrocarbon is transported through pipelines [13]. For that reason, safety operating pressure must be considered during the production, gathering, storage, and transportation of hydrocarbons through pipelines. It is known that the internal corrosion starts from brine environment, water cut (oilfield produced water, or liquid holdup), and more corrosive emulsions [12]. The internal wall of the pipeline steels and storage tanks exposed to brine reservoir can be affected by corrosive gases including hydrogen element, high salt concentrations, solids, metals, and pressure effects [96,97]. In this way, pipeline steels transporting crude oil, condensed hydrocarbons, refined hydrocarbons, natural dry gases and natural wet gases could have different corrosive media such as oxygen, carbon dioxide (CO_2_), carbonic acid (H_2_CO_3_), hydrogen sulfide (H_2_S), naphthenic acid, water cut, salts content, physicochemical properties, organic acids, among other data [12,13,44,97,169]. As the direct contact between the internal steel surface and fluid (refined hydrocarbons, oil, produced water, wet gas, dry gas, and field sludge) take place, it is interesting to research the interplay between them and the resulting corrosion initiation and propagation such as pitting. That is because pipeline internal wall are always in a susceptibility stage. It means that the propagation of the generalized and localized (pitting) corrosion damage could be affected by the internal SCC and defects (indications).

There are some standard practices that simulate the corrosive medium which is related to crude oil, dry and wet gas for internal corrosion studies [44,96,97]. A recommended standard test for SCC study is NACE Standard TM0198 [44]. In addition, different mixtures of saline water solution containing dissolved CO_2_, acid gasses (H_2_S), acetic acid, and oxygen may be prepared at laboratory level as sour environment for promoting hydrogen and CO_2_ gas for deterioration investigation [97,166]. Some physicochemical, mechanical, and electrochemical properties that have been reported to simulate the internal corrosion process and SCC for low carbon steels, mainly [27,37,38,208,209,210,211,212,213,214,215,216]. Because hydrogen gas comes from the external and internal sources, the SCC pipeline steels is an important topic which needs to be researched considering the electrolytically hydrogen charged on steel surfaces using cathodic potential. In this way, internal corrosion process can be studied where the hydrogen related degradation is highlighted using cathodic polarization [210]. However, for carbon steel corroded surfaces, the interpretation of measurements is largely restricted to elemental/chemical fingerprinting. It should be a topic of great academic and industrial interest after an experimental revision.

In field practices from oil pipelines in México, the physicochemical parameters are obtained from direct and indirect assessment, in accordance with the standard recommended practices for complementary internal SCC studies, resulting in oil API° around 4° to 22°, temperatures around 20 °C to 180 °C, oil viscosity around 0.105 kg/ms, oil density from 820 kg/cm^3^ to 920 kg/cm^3^, pressure around 20 kg/cm^2^ to 70 kg/cm^2^ (284 lb/in^2^ to 995 lb/in^2^), volumetric flow about 0.441 m^3^/s, sediments about 0.11 wt % (0.5 wt % maximum), velocity from 0.5 to 11 mph (0.22 m/s to 4.9 m/s), pH around 4 to 11, water cut around 20% volume to 70% volume, water in hydrocarbon 0.1% volume (0.5% volume maximum), H_2_, CO_2_, O_2_ around 10,000 ppb, organic acid about 0.75 mg KOH/g (0.28 maximum), H_2_S about 10,000 ppm, metals about 300 ppm, sulphates about 1688 ppm, chlorides (up to 60,000 ppm), Fe^2+^ (up to 100 ppm), hydrocarbon quality, chemical treatments (up to 125 ppm dosage), salt content about 57 pounds/1000 barrels per day (50 pounds/1000 barrels per day maximum), mass loss (up to 33 wt %), different types of emulsions, metal such as vanadium about 253 ppm (270 ppm maximum). In the case of mechanical properties, YS around 396 to 600 MPa, and UTS around 500 to 900 MPa may be achieved. In microbiological properties, aerobes, facultative, and anaerobes could be assessed, as well as Desulfobacterales, Desulfovibrionales, and Desulfuromonadales, and other sulphate reducing bacteria (SRB, such as *Clostridia*), Fe (III) reducing bacteria and acid producing bacteria also could be highlighted [205,206].

## 5. Comparison of Parameters Studied and Reported in the Literature

For the purpose of the comparison, some figures were made using data acquired from parameters studied and reported in the literature regarding the external and internal SCC of carbon steel pipelines, which were registered in Table 2, Table 3, Table 4, Table 5 and Table 6 for external properties, and Table 8, Table 9 and Table 10 for internal properties. As a result, Figure 4, Figure 5 and Figure 6 were plotted and an analysis of the more representative external environments affecting operating pipelines to develop SCC was addressed [4,6,7,11,12,13,17,37,210,211]. Figure 4a shows the external parameters regarding physicochemical and electrochemical behavior [4,7,11,17]. According to this figure, it is possible to note that the moisture and redox potential values were about 22.5 to 52.3 wt % and 265 to 373 mV, respectively, which corresponded to near acidic to alkaline soil (pH around 6.0 to 8.1). In addition, Figure 4b shows that the natural corrosion potentials (E_corr_ values) for X52, X70, X80, and X100 steels were around −400 to −1300 mV vs. Cu/CuSO_4_ electrode, when pH value was near acidic to alkaline solution. The natural potentials obtained on X70 steel at different pH were more studied and documented than the rest of carbon steels. According to pH, it is important to consider that how is the atomic hydrogen formed in the OD and how is it formed in the ID maybe the key in the similarities and differences between the external and internal cracking susceptibility, and it should be addressed. It is important to note that about the paper is it has a lot of corrosion science papers and few corrosion papers. Corrosion has many, perhaps superior publications, in this area on this topic (year: 1960–2015, total articles: 5530, SCC articles: 1050, SSRT articles: 370) [233]. However, in its current form, it seem that no external and internal SCC affected by the physicochemical, mechanical, and electrochemical data is highlighted, with particular interest in carbon steel pipelines transporting hydrocarbons.

Figure 5 suggests that the UTS values that concern the external parameters increased as YS is increased, too; while EL ranges from 27% to 40% were obtained, which corresponded to YS around 400 to 600 MPa [4,7,11,12,13,37,210,211]. As regards internal corrosion, UTS values were around 425 to 1050 MPa when YS were about 430 up to 550 MPa, while EL values have been estimated as 28% to 44% in the range of YS from 430 up to 549 MPa. In addition, the external parameters related to UTS and EL versus YS were minor in comparison with those data obtained in the literature regarding internal SCC; indicating that more susceptibility to internal SCC could occur. In comparison with the mechanical–electrochemical–physicochemical parameters, Figure 6 shows that the pH in solution is an important parameter affecting the mechanical properties; which was lower for internal corrosion where atomic hydrogen production may occur [4,7,11,12,13,17,37,210,211]. Thus, acidic media or microorganisms must be the dominant internal factors to develop SCC affecting the operating pipelines. It is not surprising that pitting corrosion or active metal dissolution could be used to ensure the SCC occurrence toward high corrosion resulting in local cracks and severe localized corrosion of the internal pipeline. The presence of water on both sides (OD and ID) of the steel is the key reason for corrosion. However, the corrosion could be increased depending on the electrolyte type into the water. Thus, high corrosion rate attributed to significant mass loss is expected.

## 6. Recommendations

First, the topic of the reviewed manuscripts can be relevant and of great importance for parameters affecting the external and internal SCC; however, some recommendations are conferred:


**External SCC**
Edaphology studies should be used to ensure a more mechanistic investigation, and a fundamental role during cathodic protection as a function of seasonal changes because natural media (growth and decomposition of plants).It is important to point out that, the original mechanical properties of the steel specimens exposed to soils could cause a reduction of the mechanical properties after electrochemical tests (carbon steels exposed to corrosive media).All of the assessment scenarios, the environments of operating pipelines including types of soils, moisture content, pH, and redox potential have been characterized, but with some limitations related to prepared soil solutions. The change of environment in each type of soil should be removed or controlled by a suitable ore such as Zeolite.Corrosion rate studies on pipeline steels immersed in natural soil have become important in materials research in the electrochemical and geological fields, mainly.Take care of the cathodic protection systems applied from winter until the beginning of the dry season (spring) when the water content in soils begin to evaporate, as the level of the cathodic protection on pipelines exposed to different types of soils may change with the moisture content.In field conditions, soil type, pH and moisture content for natural soils are three important parameters that should be measured for possible external SCC in buried pipelines. This is because representative soil solutions will always offer saturated conditions; thus, they will not always simulate the natural modification of the water content in soils, and they depend on geographical site of the buried pipeline.SCC for API steels exposed to marine environments (seabed) should be studied.When a pipeline is designed, not only is it necessary to consider the internal fluid which is transported, but the external media concerning to SCC must also be considered.



**Internal SCC**
The environments of operating pipelines (including types of extracted oilfield produced water, heavy and extra-heavy crude oil from oilfield, temperatures, and pH, mainly) have been characterized, but with some limitations related to prepared electrolytes in accordance with standard recommended practices for internal SCC studies.In some cases, internal SCC could be activated by operating pressure and temperature during the production, gathering, storage and transportation of hydrocarbons through pipelines.Different manufactured steels with different exposure areas (geometrical areas) were reported. Therefore, mechanical–electrochemical mechanisms related to SCC depending on steel surface conditions where a specific active (anode-cathode) site are expected.Small modifications of the physicochemical properties of the solution significantly could change the interactions between the active site of the steel and electroactive species.Weld heat affected zones in pipeline steels could cause severe modifications of the microstructure and possible SCC, and it depends on the internal electrolyte influenced.SCC damage such as oblique, circumferential, and not oblique/circumferential-neither oblique nor circumferential stacking of axial cracks should be considered in the laboratory. This is because these types of cracking can be found in the field.Studies about I-SCC must address the microstructure in each API steel grade.Complementary studies about I-SCC considering higher contents of H_2_S in the electrolytes and high temperature and pressure are necessary.


## 7. Conclusions

In this work, an experimental review of innumerable scientific articles that concern external and internal SCC on carbon steels was documented, and some conclusions are headlined as a follow:To avoid E-SCC and I-SCC in carbon steels three important parameters are recommended: (1) removal or control of the corrosive electrolytes in each change of environment, (2) use a suitable carbon steel with high quality, and (3) select the appropriate physical (coating), electrochemical (cathodic protection), and chemical (biocide or corrosion inhibitor) treatment applications.The evolution of the knowledge on stress corrosion cracking was taken from the fatigue corrosion process.In oil and gas industry, the SCC, HIC, and SSC processes are attributed to hydrogen (proton) becoming corrosive gas, that comes from the combining external and internal environmental sources such as sulfide compounds, acidic soils, acidic atmospheric compounds, HCl, H_2_SO_4_, H_2_O, naphthenic acid, organic acids (acetic acid, mainly), among other data.Hydrogen entrance into the pipeline is the main cause of cracking and fracture, either from the outside diameter (OD) or the inside diameter (ID).The atomic hydrogen generated on both surfaces of the steel is the one that diffuses into the steel and may reduce its mechanical integrity by generating cracks.Corrosion damage for pipelines transporting hydrocarbons depends on the electrolyte type (internal and external) which is commonly found in a specific geological and geographical site.Not all evidence for electrolytes may be expected from experimental tests at laboratory level. This is because more physicochemical modifications could be expected in natural electrolytes.In some cases, external SCC could be initiated by atomic hydrogen that comes from over-voltage during cathodic protection process.External and internal SCC is stronger influenced by bicarbonates, CO_2_, carbonic acid (H_2_CO_3_), and oxygen. However, the presence of water on both OD and ID of the steel is the role for corrosivity, susceptibility, and tensile stress on the E-SCC and I-SCC science. Therefore, oxygen, carbonates/bicarbonates concentrations, heavy metals, and sulfides and high strength are not important in the absence of water.Due to the production of corrosive microbial metabolites by different bacteria, it is important to carry out classical and molecular microbiological studies to determine, assess, and evaluate the type of microbial community found in the pipeline site. This information can be related to the chemical analysis of the soil—if an external corrosion has been determined in the SCC study—or to the fluid inside the pipeline—in the case of internal corrosion evaluation results being positive and the consequent evidence of the SCC phenomenon confirmed. Furthermore, it will depend upon the behaviors or electrochemical profiles that are carried out with the microorganisms detected in the site of the pipeline.Different types of external SCC cracks were reviewed from previous works such as micro-cracks, sharp intergranular cracks, initiation of cracks, intergranular cracks, stress corrosion cracks, intergranular and trans-granular cracks narrow and shallow cracks, “linked” and parallel cracks.For the internal corrosion the following SCC cracks were reviewed: secondary cleavage cracks, nucleated cracks, intergranular cracks, longitudinally orientated cracks, shallow cracks, circumferential cracks, stress corrosion cracks, deepest cracks, orientated cracks, inter-dendritic cracks, shallow cracks, small and large cracks, yielding stress corrosion cracks, finger-like cracks emerging from pits, propagated cracks, propagated trans-granular cracks, and individual cracks.

## Figures and Tables

**Figure 1 materials-13-05771-f001:**
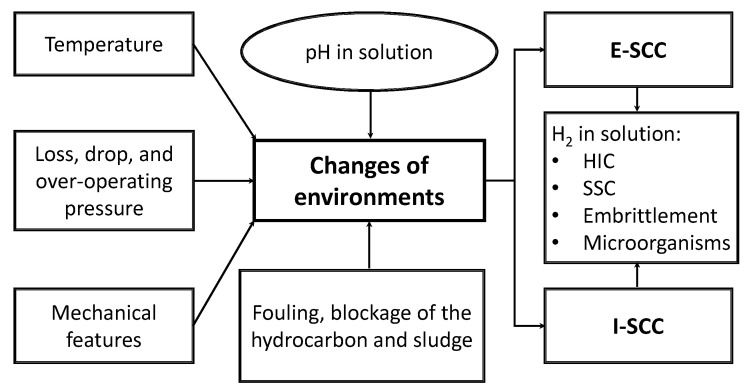
Main factors required to develop stress corrosion cracking (SCC).

**Figure 2 materials-13-05771-f002:**
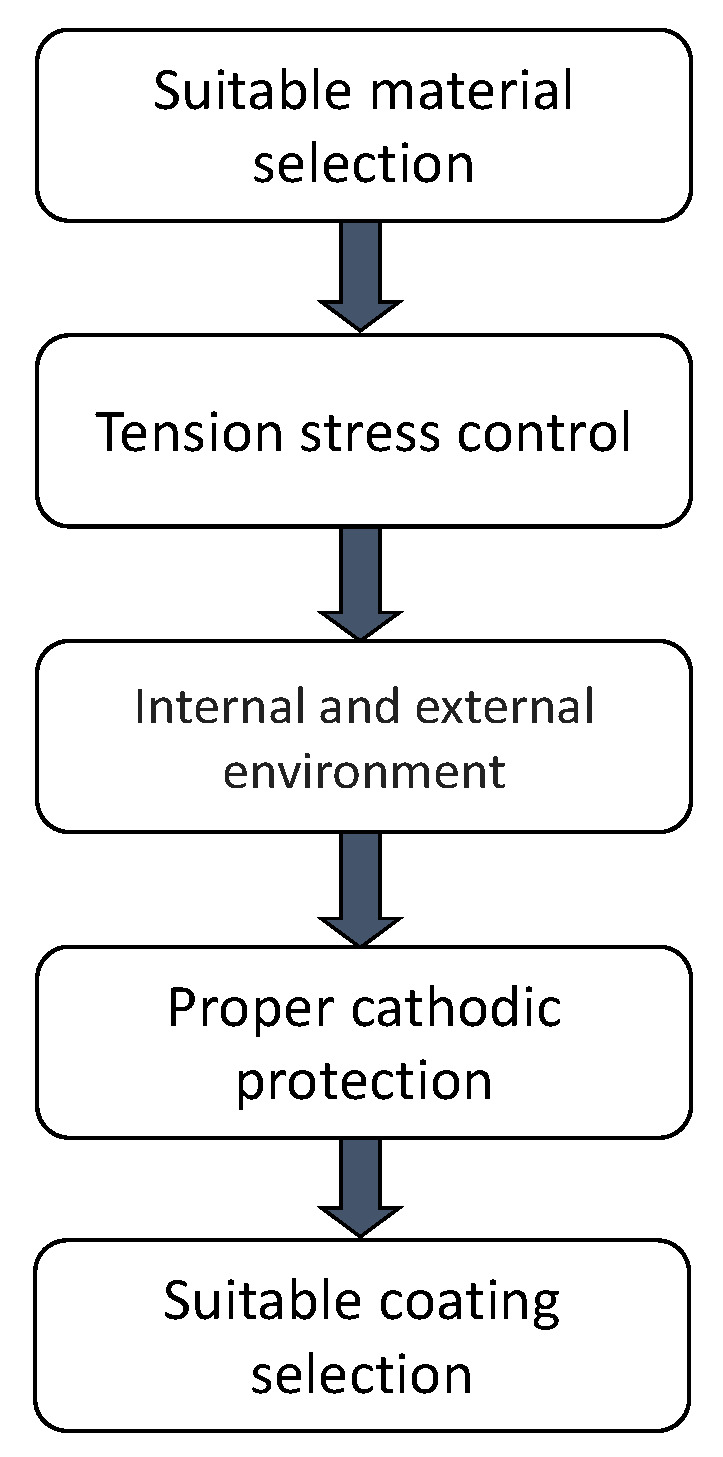
Factors that must be considered to prevent SCC development in pipelines.

**Figure 3 materials-13-05771-f003:**
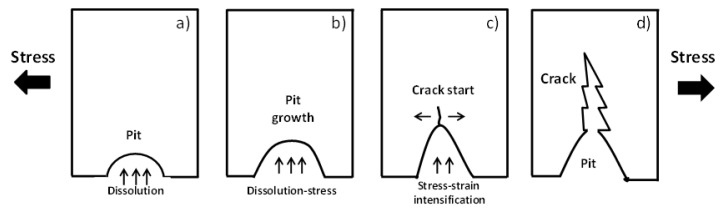
Schematic representation of the pit to crack transition [59]. (**a**) pit initiation, (**b**) pit growth, (**c**) crack start, and (**d**) final crack.

**Figure 4 materials-13-05771-f004:**
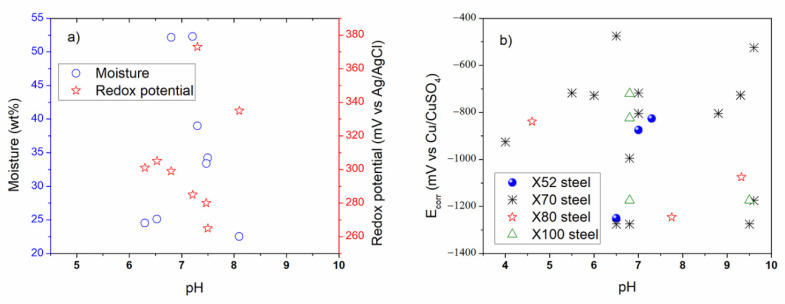
External corrosion analysis based on the reviewed (**a**) physicochemical properties, and (**b**) physicochemical and electrochemical properties.

**Figure 5 materials-13-05771-f005:**
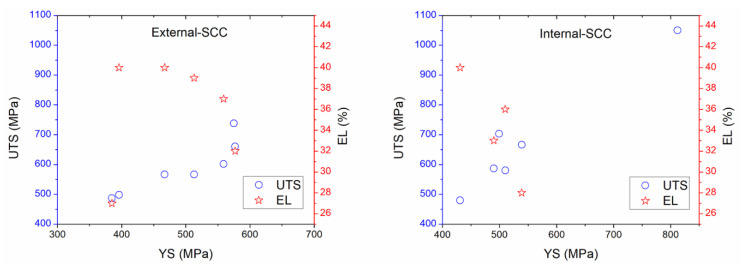
Comparison of the external- and internal-stress corrosion cracking ((E-SCC) and (I-SCC)) in respect to mechanical properties based on the reviewed experimental data.

**Figure 6 materials-13-05771-f006:**
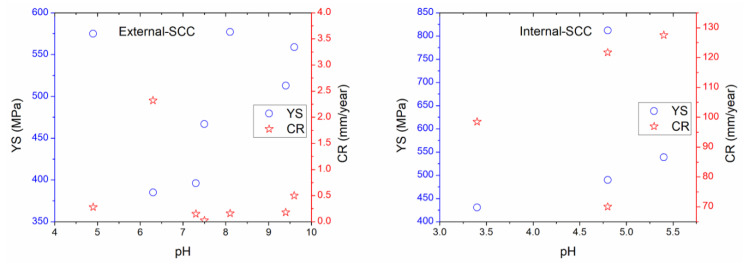
Comparison of the E-SCC and I-SCC behavior based on mechanical–electrochemical–physicochemical properties.

**Table 1 materials-13-05771-t001:** Main external environments affecting operating pipelines to develop SCC.

Factors	Parameters					Ref.
Chemical	pH	Carbonates	Organic acids	Ions	Mass loss	[43]
Electrical	Soil resistance	Potential	Current	Redox potential	Resistivity	[43]
Physical	Soil, seawater, seabed, atmospheric	Moisture	Temperature	Coating	Exposure time	[9,10,195]
Biological	Organic mater	Nutrients	Biofilm formation	Bacteria	Fungi	[197]

**Table 2 materials-13-05771-t002:** Reviewed physicochemical properties for external corrosion studies.

Steel	Coating	Electrolyte	pH	Moisture (wt %)	Redox Potential (mV vs. Ag/AgCl)	T (°C)	Conductivity (mS/cm)	Ref.
X70	FBE	Na_2_CO_3_+NaHCO_3_ + NaCl	9.4	-	-	21	61.5	[4]
X65	FBE	Na_2_CO_3_ + NaHCO_3_ + NaCl	9.6	-	-	22	21.8	[6]
X70	Polyethylene	NS4	6.8	-	-	22	-	[7]
X52	-	Sand soil	-	1.5/25	-	20	-	[9]
X52	-	Clay soil	-	25/39	-	20	-	[9]
X52	-	Marshy soil	-	25/53	-	20	-	[9]
X60	Damaged viscoelastic polymer	Sand soil	6.53	25.1	305	20	-	[9]
X60		Clay soil	7.47	33.4	280	20	-	[9]
X60		Clay-Silt soil	7.21	52.3	285	20	-	[9]
X52/X80	-	Clay soil	7.3	5.1/39.7	50/373	20	-	[11]
X70	-	NS4	6.3/6.5	-	-	22	-	[14]
X80	-	Soil solution	4.0	-	-	20	-	[15]
X80	-	Red soil	4.9	-	-	20	0.018	[17]
X70	-	Red soil	4.41	-	-	22	0.018	[18]
X52	-	Na_2_SO_4_	6.2/8.0	-	-	25/80	-	[19]
X52	-	Calcareous soil	8.1	22.5	335	20		[59]
X70	PMMA	Sand	9.6	-	-	20	-	[5,124]

**Table 3 materials-13-05771-t003:** Continued reviewed physicochemical properties for external corrosion studies.

Steel	Coating	Electrolyte	pH	T (°C)	Conductivity (μS/cm)	Ref.
X70	-	Na_2_CO_3_ + NaHCO_3_ + NaCl	9.6	-	-	[16]
X80	-	Meadow soil + SRB	7.75	-	-	[23]
X70	-	NaCl + Na_2_SO_4_ + NaHCO_3_ + SRB	8.8	4/121	-	[24]
X80	-	Acidic soil solution	4.6	-	212	[36]
X80	PMMA	NS4	-	-	950	[124]
X70	-	Soil-extracted	4.0	-	-	[133]
X52	-	NS4	6.8	27	-	[159]
X65	-	NOTW	7.1	-	-	[160]
X70	-	Na_2_CO_3_ + NaHCO_3_ + NaCl	9.6	22	-	[162]
X70	-	NS4	6.3	22	-	[163]
X70	-	NS4	6.5	22	-	[164]
X70	-	Na_2_CO_3_ + NaHCO_3_	9.3	50	-	[165]
X100	-	NS4	6.8	22	-	[198]
X80	-	Meadow soil/SRB	7.75	22	-	[199,200]
X70	-	NS4	7.0	22	-	[201]
X80	-	Na_2_CO_3_ + NaHCO_3_	9.32	-	-	[202]
S235JR	-	VMNI, wolfe’s, vitamin; *G. Sulfurreducens*	-	25	-	[203]

**Table 4 materials-13-05771-t004:** Reviewed mechanical properties for external corrosion studies.

Steel	YS (MPa)	UTS (MPa)	E (GPa)	EL (%)	Ref.
X70	650	-	-	-	[4]
X60	467	567	201	40	[9]
X52	396	498	258	40	[11]
X60	513	567	292	39	[11]
X65	559	602	298	37	[11]
X70	577	660	295	32	[11]
X80	575	738	302	23	[11]
X80	516	625	-	-	[15]
X80	581	655	-	-	[17]
X70	-	650	-	-	[18]
X52	358	455	-	-	[19]
X80	650.7	694.3	-	-	[23]
X80	581	655	-	-	[124]
X70	600	-	-	-	[163]
X100	800	870	-	-	[198]
X80	580.2	665.2	-	70	[200]
X80	560	635	-	-	[201]
S235JR	406	510	-	-	[203]

**Table 5 materials-13-05771-t005:** Reviewed electrochemical properties for external corrosion studies.

Steel	Corrosion Rate, CR (mm/Year)	Corrosion Potential, E_corr_ (mV vs. Cu/CuSO_4_)	Charge Transfer Resistance, R_ct_ (Ω·cm^2^)	Ref.
X70	-	−817	-	[4]
X70	-	−550	-	[5]
X70	-	−200	-	[7]
X60	0.00019	−951	-	[9]
X60	0.00024	−817	-	[9]
X60	0.0029	−1081	-	[9]
X52	0.15	−826	1107	[11]
X60	0.18	−831	1159	[11]
X65	0.50	−735	693	[11]
X70	0.16	−841	1042	[11]
X80	0.28	−821	1090	[11]
X52	0.025/0.079	−583/−955	17,703/11,108	[11]
X52	0.13/0.18	−315/−1000	2146/3728	[11]
X52	0.22/0.38	−817/−949	1886/2659	[11]
X70	-	−905/−995	-	[14]
X80	2.32/4.64	−665/−695	34/35	[15]
X70	0.01	−464	10,409	[16]
X80	-	−775	-	[17]
X80	-	−775	-	[18]
X52	-	−875	-	[19]
X80	-	-	12,000	[23]
X70	-	−875	-	[163]
X70	-	−835	1700	[164]
X100	-	-	2742/2843	[198]

**Table 6 materials-13-05771-t006:** Continued reviewed electrochemical properties for external corrosion studies.

Steel	Corrosion Rate, CR (mm/Year)	Corrosion Potential, E_corr_ (mV vs. Cu/CuSO_4_)	Charge Transfer Resistance, R_ct_ (Ω·cm^2^)	Ref.
X70	-	−850/−1075	909/3791	[24]
X80	0.006	−835	40,000	[36]
X80	-	-	966/2600	[124]
X70	-	−675	-	[133]
X52	-	−960	-	[159]
X70	-	−825	500/600	[162]
X70	0.9	−947	2300	[165]
X70	-	-	153/215	[201]
X80	-	−1075	-	[202]
S235JR	0.02/0.2	−675	-	[203]

**Table 7 materials-13-05771-t007:** Main internal environments affecting operating pipelines to develop SCC.

Factors	Parameters						Ref.
Electrical	Conductivity	Redox potential	-	-	-	-	[43]
Physical	Oil API°	Oil viscosity	Oil density	Pressure, temperature	Volumetric flow	Sediments	[194]
Chemical	pH	Water cut, water associated with hydrocarbons	CO_2_, O_2_	Organic acid, H_2_S, H_2_	Metals	Sulfates	[194]
Tribological	Friction	Wear	Lubrication	Erosion	-	-	[196] [207]
Biological	Bacteria	*Desulfobacterales Desulfovibrionales*		SRB	Acid producing bacteria	-	[1,197,205,206]

Note: Oil API° is the API gravity of hydrocarbons (in degrees).

**Table 8 materials-13-05771-t008:** Reviewed physicochemical properties for internal corrosion studies.

Steel	Electrolyte	pH	T (°C)	Ref.
X80	0.1 M Na_2_SO_4_ + CO_2_; 0.1 M NaHCO_3_ + 0.17 M NaCl	4.2/8.4	25	[37]
Inconel 625/X60	NACE Standard TM0177 test solution A	-	-	[38]
X52	1 M H_2_SO_4_	-	-	[208]
Fe-C alloy	8.5 M NaOH	13.2	100	[209]
X70	3% NaCl + H_2_ (1.5–2.5 ppm)	-	4/80	[210]
X52	NACE Standard TM0284; NACE Standard TM0177	3.4/5.4	200	[211]
X65				
X70				
X70	0.122 g/L KCl + 0.483 g/L NaHCO_3_, 0.181 g/L CaCl_2_·2H_2_O + 0.131 g/L MgSO_4_·7H_2_O; or sodium chloride (NaCl), CaCl_2_, and Na_2_SO_4_ at various concentrations and adding CO_2_ gas.	6.5/11.9	20/40	[212]
X65	200-proof ethanol, anhydrous methanol, glacial acetic acid and nano-pure deionized water.	7.6/7.7	-	[213]
X70	Dilute brine bubbled with 10% CO_2_ and dilute brine with 1% H_2_S	4.9	30	[214]
X65	3.5 wt % NaCl with different H_2_S: CO_2_ gas ratios added	3.8/4.4	30/80	[215]
X70	1 N Sodium carbonate plus 1 N Sodium bicarbonate	-	75	[216]

**Table 9 materials-13-05771-t009:** Reviewed mechanical properties for internal corrosion studies.

Steel	YS (MPa)	UTS (MPa)	EL (%)	Ref.
X60	468/499	703/520	-	[38]
X70	812	1050	-	[210]
X52	431	480	40	[211]
X65	490	587	33	[211]
X70	539	666	28	[211]
X65	690	810	-	[213]
X70	510	580	36	[216]

**Table 10 materials-13-05771-t010:** Reviewed electrochemical properties for internal corrosion studies.

Steel	Corrosion Rate, CR (mm/Year)	Corrosion Potential, E_corr_ (mV vs. SCE)	Charge Transfer Resistance, R_ct_ (Ω·cm^2^)	Ref.
X80	1.75/0.014	−660/−634	3000	[37]
X60	0.01/0.69	-	-	[38]
X65	0.5/2.0	−864/−784	-	[200]
Fe-C alloy	0.81/1.27	-	-	[209]
X52	98.5	−680	-	[211]
X65	121.7	−680	-	[211]
X70	127.5	−690	-	[211]
X70	-	−600/−700	-	[212]
X70	-	−650	-	[216]

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
