# Peer review of "Analysis of the Physicochemical, Mechanical, and Electrochemical Parameters and Their Impact on the Internal and External SCC of Carbon Steel Pipelines"

_materials, 2020, doi:10.3390/ma13245771_

Round 1

Reviewer 1 Report

Dear authors;

really the work is very well documented and of course it shows that there is a lot of bibliographic work in it.
However, the article seems more suitable for the introduction of a doctoral thesis than for a scientific article.

It is too long and with some basic explanations.

My recommendation is that, as far as possible, try to reduce the text while preserving the most substantial part.

The abstract is too long. From my point of view, section 2.1 should not be written interrogatively.

Subsection 2.6.3 is quite long with explanations that can be found in any textbook. should be reduced.

Author Response

Response 1: The main manuscript was reduced the similarity.

Reviewer 2 Report

Comments to the Authors:
The authors of this paper present an interesting review regarding the  stress corrosion cracking phenomena.

More specifically, this work aims to present a review of a brief experimental summary and a comparison of physicochemical, mechanical, and electrochemical data affecting external and internal stress corrosion cracking in carbon steel pipelines exposed to corrosive media.

 It may first be mentioned that, although the abstract is well written, it is a bit long. Thus it could be reduced by the authors.

 A large number of basic standards in the area are reported in this work, while the number of references (more than two hundred) is satisfactory. It is a comprehensive and critical review, interesting for the readers and, especially, for the newcomers into the field.

 It is a well written review; the reported data found in different previously published works are compared and discussed, the recommendations provided by the authors are interesting, while the results support the authors’ conclusions.

 (A minor misprint: page 28, line 1096, substitute "paremeters" for “parameters”)

 This work can also be a reference point for future research studies in this field. Thus this article may be published."

Author Response

Response: the main manuscript was significantly reduced.

Reviewer 3 Report

This is an extensive and high-quality overview study of corrosion effects on carbon steel pipes.

The article is current, very well processed.

The testing procedures focus on those used in the North American petroleum industry (Mexico, USA and Canada).

An overview of methods for testing pipe materials and testing standards is presented. The paper is precisely processed.

The authors present a wide range of issues, drawing on 233 sources of literature. The length of 42 pages is quite large.

I recommend reducing the abstract to the prescribed number of words. in accordance with the instructions.

Table. 2 a Table. 7 check and adjust formatting

Check of subscripts (in formulas, chemical compounds) in the paper.

Author Response

Response: Tables and formulas were corrected.
